# Repeated Etching Cycles of Resin Infiltration up to Nine Cycles on Demineralized Enamel: Surface Roughness and Esthetic Outcomes—In Vitro Study

**DOI:** 10.3390/children10071148

**Published:** 2023-06-30

**Authors:** Dayang Fadzlina Abang Ibrahim, Noren Nor Hasmun, Yih Miin Liew, Annapurny Venkiteswaran

**Affiliations:** 1Centre of Paediatric Dentistry and Orthodontic Studies, Faculty of Dentistry, Universiti Teknologi MARA (UiTM) Campus Sungai Buloh, Sungai Buloh 47000, Malaysia; dayangfadzlina3@gmail.com; 2Department of Oral Sciences, Faculty of Dentistry, University of Otago, P.O. Box 56, Dunedin 9054, New Zealand; noren.hasmun@otago.ac.nz; 3Department of Biomedical Engineering, Faculty of Engineering, Universiti Malaya, Kuala Lumpur 50603, Malaysia; liewym@um.edu.my

**Keywords:** resin infiltration, hydrochloric acid, surface roughness, esthetics, demineralized enamel

## Abstract

Resin infiltration (RI) is used to mask enamel opacities. There are three recommended etching cycles. However, anecdotal evidence suggests that favorable esthetics outcomes can be obtained by increasing the etching cycles. This study aimed to evaluate the effects of repeated etching cycles during RI application on esthetic changes and surface roughness of demineralized enamel at multiple treatment stages. Artificial demineralization was prepared on the buccal surface of ninety sound extracted premolars. The teeth were divided into nine groups (n = 10); with each consecutive group having one additional etching cycle up to nine etching cycles. Resin infiltrant was performed twice, first for 3 min (Resin 1) and again for 1 min (Resin 2). Surface roughness and esthetic changes were assessed using a profilometer (Ambios XP-200) and Minolta spectrophotometer, respectively, at baseline (sound enamel), etching, resin 1, resin 2, 7 days, and 28 days post resin applications. Data were analyzed with two-way ANOVA (*p* < 0.05). There was a significant interaction between the different stages and various groups of etching cycles on surface roughness, F(48, 126) = 3.48, *p* < 0.001. There was a significant interaction between the different stages and various groups of etching cycles on color changes, F(4, 126) = 1.177, *p* = 0.045. The surface roughness of demineralized enamel infiltrated with RI was less than that of sound enamel (baseline). There is a significant difference in color changes between resin 1 and resin 2 (*p* < 0.05). After five etching cycles, RI improved the esthetic of the color of teeth similar to the baseline. Surface roughness and color changes remained constant for 28 days. RI can be considered an effective and predictable treatment option for the restoration of early enamel lesions owing to its better surface characteristics and reliable masking effects. The color stability and surface roughness stay unaltered for up to 28 days.

## 1. Introduction

In recent years, the management of dental caries has dramatically shifted from a traditional restorative treatment approach to a more preventive approach, namely non-invasion or minimally invasive dentistry [1]. Early enamel caries are characterized by mineral loss in the lesion’s body, which results in increased visual enamel opacity due to a change in the refractive index of the affected region [2]. The noninvasive treatment of early enamel caries, such as remineralization with fluoride and casein phosphopeptide-amorphous calcium phosphate (CPP-ACP) or the use of therapeutic sealants for occlusal lesions, has received a great deal of attention. Fluoride and CPP-ACP are essential for the remineralization of superficial white spot lesions (WSL) [3]. This approach is not always successful, as it requires patient compliance and a change in detrimental behaviors, and many patients abandon treatment before completion [4].

In 2009, resin infiltration (RI) was developed in Hamburg, Germany and introduced as a microinvasive treatment option for enamel lesions extending into the outer third of non-cavitated dentine [5]. ICON^TM^, which stands for “Infiltration concept,” is the brand name for a RI product that comes in kits for the proximal and vestibular surfaces. It provides good access to the lesion site and was primarily designed to treat early proximal caries lesions without removing healthy enamel [6]. This product contains 15% hydrochloric acid (HCl) as an etchant, ethanol as a dehydrating agent, and resin infiltrant made of triethylene glycol dimethacrylate (TEGDMA) [7,8]. RI technique entails etching the tooth lesion surface with 15% HCl for two minutes, adequately desiccating the tooth, and injecting a low-viscosity resin into the demineralized enamel’s intercrystallite gaps, followed by light-curing the resin material [7,8]. The low viscosity RI has a tendency to obstruct enamel porosity that serves as diffusion channels for acids and dissolved minerals to penetrate the demineralized enamel. Moreover, since the infiltrated resin material has a refractive index comparable to that of tooth hydroxyapatite, the optical manifestations of the affected enamel can mimic those of a healthy tooth [8,9]. Thus, RI is one of the techniques used to mask enamel opacities including WSL [6,7]. RI has been shown to be a convincing alternative to topical fluoride therapy for the management of demineralized enamel [6]. RI offers benefits such as mechanical stabilization of the demineralized lesion, preservation of sound hard substance, permanent occlusion of the superficial micropores and cavities, arrest of lesion progression, minimized risk of secondary caries and delay of restorative intervention for longer periods [6,7]. Non-cavitated enamel caries and dentin caries that have not advanced beyond the first third of the dentin, lesions produced by molar incisor hypomineralization, hypoplasia discolorations caused by acute trauma and fluorosis are indications for RI [8].

A systematic review reported that RI is effective for concealing enamel discoloration [8,9]. Despite following the manufacturer’s instructions, the esthetic outcomes are unpredictable, hence some clinicians are still hesitant to use this material [8]. Anecdotal evidence suggested that esthetic outcomes of demineralized enamel treated with RI can be enhanced, by increasing the number of etching cycles prior to RI [9]. As a result, some clinicians have altered the procedure for applying RI in order to maximize its efficacy in concealing the opacity caused by demineralized enamel [10]. These alterations include exceeding the manufacturer’s recommended number of three etching cycles or combining this intervention with another therapy, such as microabrasion or bleaching [10]. This has been shown to be successful in a clinical setting, but no research has evaluated the number of etching cycles needed to achieve the optimum esthetic results while minimizing the enamel loss and surface roughness following etchant placement on demineralized enamel.

This would be of therapeutic interest to a clinician who must achieve optimal results in masking the demineralized enamel. This is the first study that conducted the number of the etching cycle to determine the best esthetic outcomes of the RI on demineralized enamel. Also, the effect of the surface roughness following repeated etching cycles was determined in this following study. It is important to know that a flawlessly smooth surface assures a shine, which enhances the overall visual acceptability of tooth color [11,12].

As a result, it is possible that a significant number of etching cycles are required to improve the surface properties and color stability of RI. Repeated etching cycles before the RI procedure may improve penetration, retention, and surface roughness of the lesions. However, the effect of repeated etching cycles on the surface roughness of demineralized lesions intended for resin penetration has received little attention to date. In light of the aforementioned research problem, the purpose of this study was to determine the surface roughness and to assess the effectiveness of RI in masking WSL following repeated etching cycles. We hypothesized that there was no significant difference in surface roughness and esthetic changes following a repeated etching cycle application using 15% HCl and following a repeated resin infiltrant application.

## 2. Materials and Methods

### 2.1. Study Design and Sample Preparation

The approval to conduct the study was granted by The Research Ethics Committee of Universiti Teknologi Mara (UiTM) with the approval code REC/08/2020 (MR/220, 15 September 2020). In this in vitro study, power analysis (using Axum 7 (Mathsoft, Cambridge, MA, USA)) was carried out with an alpha of 0.05 and a power of 0.9 (sigma 1 = 10.00, sigma 2 = 11.35; mean 1 = 35.95, mean 2 = 49.37) which resulted in a sample size of 90 [13]. The inclusion criteria include sound premolar tooth extracted for orthodontic treatment with intact buccal surface. Teeth with caries, developmental defects of enamel (Amelogenesis imperfecta, Dentinogenesis imperfecta, fluorosis, molar incisor hypomineralization) or presence of restoration were excluded. All teeth were coated with nail varnish (Revlon, New York, NY, USA) leaving a 4 mm × 4 mm square which was treated as region of interest (ROI) on the midbuccal surface of experimental teeth. Only this window was exposed for the induction of artificial lesions. Teeth were stored at room temperature in a 0.1% thymol solution (BDH Chemicals Ltd., Poole, England) to prevent dehydration and additional microbial growth. The specimen was randomly selected and further divided into nine groups (A-I). Each group consists of 10 teeth samples. The difference between each group is the number of etching cycles. For example, teeth in Group A were undergo one etching cycle while teeth in Group B were treated with an additional etching cycle (2 cycles). Each group was subjected to acid etching application (15% HCl; ICON^TM^ Etch) followed by resin infiltrant treatment application (ICON^TM^ Infiltrant). The infiltrant was allowed to penetrate for 3 min, air-dried and light cured for 40 s (Resin 1). The lesion was infiltrated for an additional 1 min and light cured for 40 s (Resin 2). Then, the resin was polished with 11,000 rpm fine and superfine polishing disc (Soft-Lex, 3M). Figure 1 showed the workflow of the study.

### 2.2. Artificial Demineralization

All teeth were coated with nail varnish (Revlon, New York, NY, USA) leaving a 4 mm × 4 mm square which was treated as a ROI on the midbuccal surface of experimental teeth. Artificial WSL were created in the uncoated areas by storing the teeth in a 5 L demineralizing solution containing 2.2 mM calcium chloride, 2.2 mM monopotassium phosphate, 0.05 mM acetic acid, and 6 μm metylhydroxydiphospho-nate (pH 4.4) at 37 °C for seven days [14]. The acidity of the solution was checked daily with a pH meter to ensure the solution is stable or near pH 4.4, and the tooth examines for the development of WSL [14]. The teeth were examined visually using International Caries Detection and Assessment System (ICDAS) codes to ensure that only teeth with codes 1 and 2 (early stage of dental caries) were included. Each tooth was mounted in an acrylic resin to ease the handling for the analysis. After demineralization, the teeth were removed from the demineralizing solution, rinsed with deionized water, and then desiccated.

### 2.3. pH Cycling

Following artificial demineralization, the teeth underwent pH cycling. Using pH cycling, an artificial caries lesion was created to simulate the intraoral situation characterized by alternating demineralization and remineralization processes. The pH cycling model utilized in this study, known as the modified Featherstone model [15], consisted of immersing all samples for 6 h in a demineralization solution containing 2.0 mM calcium nitrate tetrahydrate, 2.0 mM monopotassium phosphate, 75.0 mM acetic acid and sodium hydroxide. After demineralization, all samples were immersed for 17.5 h in a remineralization solution containing 1.5 mM calcium nitrate tetrahydrate, 0.9 mM monopotassium phosphate, 130.0 mM potassium chloride, 20.0 mM sodium cacodylate, and hydrochloric acid at a pH of 7.0. The demineralization and remineralization procedures were alternated and repeated for up to 28 days [16]. The profilometer and spectrophotometer scans were performed prior to pH cycling (to obtain baseline measurement) and at various stages (post-etching, resin 1 and resin 2, 7 days, and 28 days post resin applications).

### 2.4. Resin Infiltration Applications

RI was applied by the same operator (DFA) to the ROI. First, 15% HCl (ICON^TM^ Etch) was agitated to the lesions for two minutes and rinsed with water for 30 s. After drying the lesions with oil- and water-free air, the lesions were treated with ethanol (ICON^TM^ Dry) and left to evaporate for 30 s [17]. These steps will be repeated according to which group the teeth were assigned to. For example, teeth in Group A had one etching cycle, rinsed and dried, followed by the application of resin infiltrant (ICON^TM^ Infiltrant). Teeth in Group B, on the other hand, had two etching cycles then rinsed and dried twice before resin infiltrant was applied. Resin infiltrant (ICON^TM^ Infiltrant) was performed twice, first for 3 min (Resin 1) and again for 1 min (Resin 2). Similar materials were used for resins 1 and 2, which are Resin Infiltrant (ICON^TM^ Infiltrant). For Resin 1, the resin infiltrant was allowed to penetrate for 3 min, gently air dried and light cured for 40 s (Resin 1). The lesion was infiltrated for ananother minute and light cured for 40 s (Resin 2). Finally, the resin was polished with a 11,000 rpm fine and superfine polishing disc (Soft-Lex, 3M). 

### 2.5. Surface Roughness

All specimens were scanned and analyzed for surface roughness using a high surface stylus profilometer Ambios XP-200 (Ambios Technology, Inc., Milpitas, CA, USA). The stylus profilometer consisted of a vertical measurement range of 800 µm, a scan length range of 50 mm, and a stylus tip radius of 2.5 µm. To gather the surface profile data, a total scan length of 5 mm was performed at a scan speed of 0.03 mm/s [18]. Average roughness values (Ra, µm) were determined for each specimen after three measurements undertaken at six evaluation stages (baseline, post etching, resin 1, resin 2, post resin 7 days, and post resin 28 days). 

### 2.6. Esthetics

The color of the ROI was measured with a CM-5 spectrophotometer (Konica Minolta CM-5 spectrophotometer (Minolta Co., Osaka, Japan)) with a 360–740 nm wavelength range. For the color measurements, the color difference (ΔE) was calculated using the following formula [19]:ΔE* = [(ΔL*)2 + (Δa*)2 + (Δb*)2)]1/2

The L* value indicated lightness and varies between 0 (black) and 100 (white); a* determined the amount of red (positive values) or green (negative values) and b* displayed the amount of yellow (positive values) or blue (negative values). Color measurements were taken at six time points; baseline, post etching, resin 1, resin 2, post resin 7 days, and post resin 28 days. The spectrophotometer was calibrated prior to color measurement. Then, each specimen was placed on the target mask, ensuring the tooth is as close as possible to the specimen measuring port. Measurements were repeated three times and average readings of each sample were recorded. To standardize color evaluation, the following measures were taken: standardize ambient light conditions, place specimens against a white background during all color readouts, carefully prepare samples prior to color readouts (i.e., dry the tooth surface with absorbent paper), standardize the tip of the spectrophotometer (perpendicular and pressed flush against the buccal surface of the specimen) and ensure repeatable positioning during readouts via the orientation groove. The measurements were then transferred to the computer and the data were saved. All measurements were taken by the same operator (DFA).

### 2.7. Statistical Analysis

Statistical analysis was performed using Statistical Package for the Social Sciences (SPSS) Statistics for Windows, version 25 (IBM Corp., Armonk, NY, USA). A two-way ANOVA was conducted that examined the effect of the etching cycle of different groups and different stages on surface roughness and esthetic changes with a significance level of *p* < 0.05.

## 3. Results

### 3.1. Surface Roughness

The surface of enamel appears roughest during post etching and post resin 1 infiltration for all groups (Table 1). In addition, the more cycles the enamel was etched, the higher the surface roughness. The roughness was only smoothened out during the second resin infiltrant (Resin 2). At resin 2, post resin 7 days and post resin 28 days, the surface roughness appears to be consistent. There was a significant interaction between the different stages and etching cycles on surface roughness, (F(48, 126) = [3.48], *p* < 0.001) (Refer to Table 2). The box plot demonstrates that the surface roughness increased during etching, but decreased after resin 1 and resin 2 infiltration (Figure 2). The roughness values remain unchanged 7 days and 28 days post resin application.

### 3.2. Esthetic

The ΔE appears to be reduced after resin 2 (after polishing with soflex disc) compared to post etching and resin 1 for all groups. In addition, the ΔE value becomes more constant after resin 2, post resin 7 days, and post resin 28 days (Table 3). A two-way ANOVA summary table related to the ΔE for interaction of stages and various etching groups was shown in Table 4. There was a significant interaction between the different stages and various groups of etching cycles on ΔE, (F(4, 126) = 1.177, *p* = 0.045). After etching cycles, as depicted in Figure 3, the color of the ROI becomes brighter. The color of the ROI appears comparable to the baseline after resin applications at five etching cycles. The color of the ROI does not change significantly after the progressive etching cycle at cycle six to nine.

Tukey’s post hoc analysis found that the mean value ΔE was significantly different between baseline versus resin 2, post resin 7 days, and post resin 28 days (*p*-value < 0.05). Tukey’s post hoc analysis found that the mean value ΔE was significantly different between resin 1 versus resin 2, post resin 7 days, and post resin 28 days (*p*-value < 0.05) (Refer to Table 5). However, after etching cycle five and above, no significantly different between baseline versus resin 1, resin 2, post resin 7 days, and post resin 28 days (*p*-value > 0.05). This demonstrated that after five or more etching cycles, the color of the ROI was comparable to baseline color and remained constant for 28 days after resin application.

## 4. Discussion

This in vitro study assessed the surface roughness and esthetic changes resulting from RI treatment for up to 28 days. According to the present study, RI with repeated etching cycles produces esthetically superior results at five etching cycles and can maintain color stability for up to 28 days. The surface roughness and enamel porosity increased as a result of repeated etching cycles, thus improved penetration of resin and esthetics.

RI employs the concept of infiltrating the non cavitated caries lesion with a low viscosity resin in an effort to halt the progression of the lesion without removing the tooth structure [6,7]. The low viscosity RI obstructed the enamel pores that serve as diffusion channels for acids and dissolved minerals to penetrate the demineralized lesion [6,7,9]. Moreover, the optical manifestations of the affected enamel resembled those of a sound tooth since the infiltrated resin material has a similar refractive index to that of tooth hydroxyapatite [8,9]. By modifying the lesions internal light scattering with a polymerizable low viscosity resin, enamel lesions can be concealed [9].

The presence of the hypermineralized superficial pseudointact surface layer of enamel caries has the potential to impede the capillary action of the resin [9,17]. In addition, the presence of contaminated residues of dust, water, saliva, intraoral pH and organic substance on the enamel surface may hinder resin perfusion, necessitating an etching procedure prior to RI placement [6]. As demonstrated by previous studies, 15% HCl gel for 120 s results in nearly complete eradication of the surface layer and therefore appears more suitable for the pretreatment of demineralized enamel prior to RI compared to 37% phosphoric acid [20]. Arnold et al. (2016) suggested repeatedly etching the enamel with 15% HCl to increase the depth of resin penetration, thereby enhancing the efficacy of arresting the lesion and enhancing masking ability [20]. Our results are in line with Arnold’s work.

The current study suggested that RI treatment decreases surface roughness relative to baseline similar to a recent meta-analysis by Soveral et al. (2021) [9]. This demonstrated that the application of RI can seal enamel pores and the resulting enamel surface appeared seamless [8,9]. This revealed that the enamel surface treated with RI exhibited enamel rod blockage, leaving a smooth surface with a uniform topography echoing the study by Yazkan et al. (2018) [11]. Therefore, the penetration of resin into the enamel porosity lessens the light scattering, decreases the opacities in the enamel and enhances the tooth appearance [9].

Taher et al. (2013) found no significant difference in surface roughness before and after applying RI material to enamel using a profilometer [12]. This is due to the fact that Taher et al. (2013) study was performed on sound enamel without any demineralization, therefore the experiment utilizing atomic force microscopy did not demonstrate any appreciable changes in surface roughness among various treatment approaches [12].

This study revealed that surface roughness after etching using a 15% HCl differed significantly from roughness values at baseline. The surface roughness results from the loss of minerals and could be attributable to an increase in enamel porosity caused by the dissolution of hydroxyapatite crystals in the enamel surface [21]. In addition, the duration of exposure to the 15% HCl had a significant effect on the increase in surface roughness, owing to further demineralization [22]. The current study shows that, in comparison to artificial demineralization and etching cycles, surface roughness values were much lower following RI and this finding is consistent with Aswani et al. (2019) [23]. The surface roughness was reduced since the HCl applications created microporosities which were then filled with RI [8]. As a result, after applying resin to the surface, a homogenous layer with groups of microscopic enamel grains is evenly distributed, smoothing out the enamel surface and reducing surface roughness [13,14]. The infiltration of resin restored the enamel surface to a flat and smooth and no significant changes in surface roughness value over time were seen at the retrieval period of 7 days and 28 days post resin, which echoed the findings of the previous studies [13,14,22].

We observed that surface roughness values increased with etching cycles. The etching frequency affects the roughen surface properties of surface enamel, which was supported by Harsha et al. (2022) [24]. Its magnitude grew progressively following acid application and plateaued beyond seven etching cycles. Meanwhile, Tsai et al. (2019) found that the surface roughness plateaued after the application of 37% phosphoric acid for 120 s [25]. The mechanism is unknown, and additional research is required to investigate the phenomena.

The polishing and finishing protocols after RI may also influence the reduction of surface roughness in this study. Studies advised wiping off the excess RI before polymerization since the residual thin resin layer may lead to plaque collection and thus causing decay [11,21]. As a result, the surface roughness value was lower after polishing during the stage of post RI application for one minute (resin 2) compared to the first-time post RI application for three minutes (resin 1). A restoration with reduced surface roughness offers a superior surface that inhibits the production of discoloration films and the accumulation of plaque [12,22]. The resin surface roughness affects how well cariogenic biofilms adhere to and grow on their surfaces [23,24]. Comparing to the plaque retention threshold value of 0.20 µm for surface roughness, all of the samples in this study showed a higher value, corroborating the findings of El Meligy et al. (2021) [26]. It was revealed that RI can still lead to plaque accumulation. Despite the fact that RI increases surface roughness, the polishability and surface characteristics of the resin-infiltrated enamel are superior to those of the demineralized enamel, making the resin infiltrated enamel of high quality and a viable treatment option. When the surface roughness value is more than 0.50 µm, patients could notice the surface changes [11,12]. Thus, the patients must be educated on the importance of home hygiene, professional hygiene visits, and proper diet intake in order to reduce the risk of plaque accumulation.

In terms of esthetics, based on the results of this study, RI has the effect on lowering ΔE and giving immediate esthetics improvement by restoring the natural color of enamel, which is consistent with the findings of other studies [7,8,23]. This is due to the fact that the microporosities of demineralized lesions are filled with resin as opposed to water or air [6,7]. By replacing air filled pores with a resin infiltrant that closely resembles enamel, the masking of early enamel WSL is achieved, with a reduction in light scattering and ΔE close to the baseline enamel value [7,8]. Consequently, the difference in refractive indices between micropores and enamel is diminished and lesions regain their translucency resembling the surrounding enamel [7,8,9]. Moreover, the deep penetration of resin infiltrant caused plugging of porosities within the WSL may be the cause of resistance against discoloration and improvement of the color [16,17]. The color remained constant over a period of 28 days, as evidenced by the fact that no significant variations in ΔE value over time were seen at the retrieval time points [25]. The currently available clinical data on long-term color stability measured with a spectrophotometer, indicate stable infiltrates and esthetic assimilation of color and brightness differences between infiltrated lesions and baseline healthy enamel [12,18,19]. Based on our findings, it can be confirmed that RI conceals demineralized enamel lesions immediately and effectively after five etching cycles. In addition, 28 days after a significant reduction in ΔE, it remained just slightly below the perception threshold (ΔE < 3.7) [18].

Polishing resin infiltrated lesions could enhance the stabilization of the masking effect, most likely due to decreased in the surface porosity and the eradication of the oxygen inhibition layer [11,14]. ΔE < 3.7 is regarded as a clinically acceptable color difference [18]. In this investigation, the ΔE value after RI therapy was 1.39 ± 0.09, demonstrating clinically adequate color recovery of the infiltrated lesions [25,27]. This is clinically relevant since application of RI reduced enamel opacities and regained its baseline appearance [18]. This indicates that the color difference was imperceptible to the human eye, hence demonstrating a further decrease in WSL and having a significant impact on tooth esthetics [8,9,18].

The susceptibility of resin infiltrated surface properties to degenerate and exhibit increased surface irregularity and plaque accumulation may also have an effect on other surface properties, such as the color stability of the RI [7,8]. As the surface topography of a resin infiltrated lesion becomes more irregular and abrasive, the incidence light interacting with the resin infiltrated enamel scatters more aggressively, affecting the color perception of the lesion [23]. The current study findings on ΔE and ΔL following artificial WSLs concur with prior research [7,8]. The demineralization process mimics early caries and generates subsurface porosities that fill with air and water and possess a lower refractive index than enamel. Multiple interfaces of subsurface porosity within the enamel increase light ray scattering, leading to a white optical appearance and a rise in lightness (L* value) [8,9]. Furthermore, the etching procedure may interfere with the color properties, as etching removes minerals or inorganic/organic particles from the enamel [21]. After etching teeth with 15% HCl, the teeth may take a longer time to rehydrate, causing an increase in L* at the color readout after progressive etching cycles [25,27,28]. Nevertheless, the immediate application of resin may reverse and subsequently improve the color alteration [6,25].

We found that the factors that could influence the color improvement of WSL are the number of etching cycles and the number of RI application. The present study revealed that after five rounds of etching cycles, the artificial demineralized enamel was completely masked. Thus, to optimize the esthetic outcomes of resin penetration, the procedure and time must be precisely examined and managed. Dual application (dual application occurred when the resin infiltrant was applied twice, the first time for 3 min (Resin 1) and the second time for 1 min (Resin 2)) of RI resulted in enamel color improvement comparable to the baseline and consistent with the findings of other studies [25,27]. It has been shown in the literature that dual application of the resin infiltrant reduces the polymerization shrinkage of the material after the first application [25]. After resin treatment, reduced light scattering intensity of the demineralized lesions was occured, indicating that the number of micropores in the demineralized enamel was reduced [17]. This demonstrated that the infiltration of the resin hardened enamel surface reduced tooth opacity and improved its esthetics.

Overall, these in vitro studies assessed the colorimetric stability of RI over a brief or very short period (immediate evaluation, after 7 or 28 weeks), revealing that RI was resistant to color variation and accelerated aging. There are likely two mechanisms underlying the discoloration of infiltration resin; one is intrinsically related to its primary constituent TEGDMA, which has a high degree of water sorption, and the other is extrinsically related to a less than ideal surface polish that degrades over time in the oral cavity [6,23]. To improve the surface polish and esthetic outcomes of RI, particularly when smooth surface WSL are present in the esthetic zone, additional research is required.

Additionally, the degree of enamel demineralization and the depth of the lesion have an impact on the masking effect [27,28]. The depth of a lesion is a predictor of its severity, occasionally, lesions with defined borders are more severe than those with less obvious and opalescent borders [27,28,29]. In this study, the WSL is restricted to the outer one-third of enamel; consequently, the etching cycles and esthetic results were predictable. Lesions deeper than the penetration capacity of RI may exhibit insufficient esthetics [29,30]. Thus, additional research concentrating on lesion depth and color masking of developmental enamel defects and other enamel opacities is required.

The exorbitant cost of RI is one of the limitations of this study. In addition, we acknowledged that findings from this in vitro study must be interpreted with cautious since we used artificially prepared demineralized lesions and pH cycling model to stimulate the demineralization and remineralization in intraoral environment. It is not possible to generalize the findings of the present study since they are conducted in artificial, highly controlled environments. The quantity of demineralization in each tooth may also vary, depending on the amount of fluoride they were exposed to before extraction. As a potential inclusion criterion, the provenance of the samples should be considered. These investigations are limited as we did not collect the background information of the volunteers donated their teeth for this study, such as their age, diets, and environments. As a result, the orientation of enamel prisms may have varied in between teeth and thus, responses to acid etching also varied. Thus, the obtained results need to be verified by examining human teeth in vivo to validate possibility of providing clinical recommendations.

## 5. Conclusions

RI can be considered an effective and predictable treatment option for the restoration of early enamel lesions owing to its better surface characteristics and reliable masking effects. The color stability and surface roughness stay unaltered for up to 28 days.

## Figures and Tables

**Figure 1 children-10-01148-f001:**
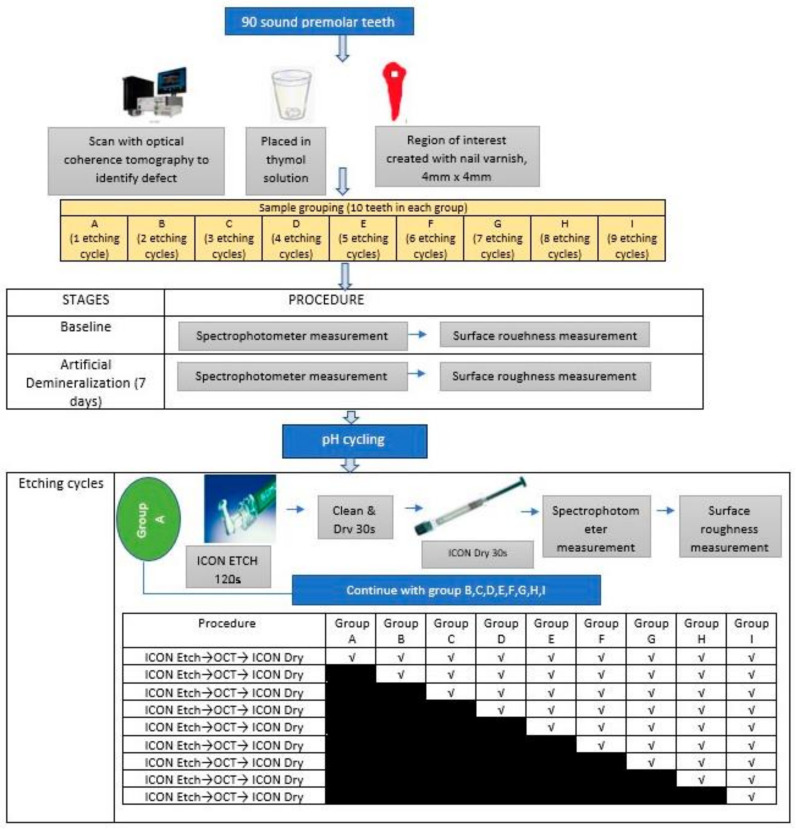
Experimental design of the study.

**Figure 2 children-10-01148-f002:**
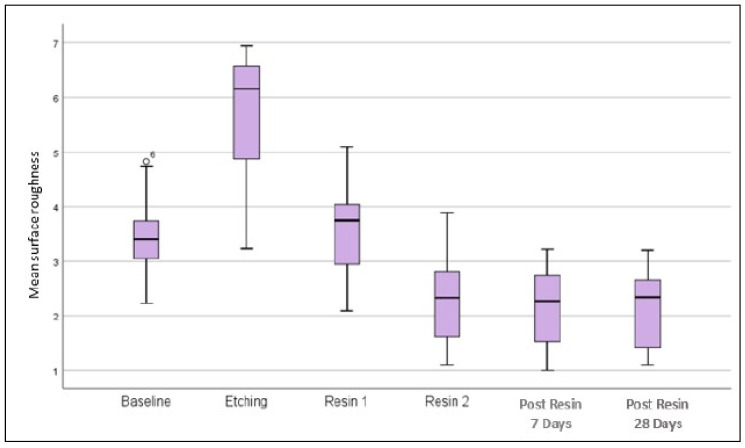
Surface roughness at various stages: baseline, etching, resin 1, resin 2, post resin 7 days, and post resin 28 days.

**Figure 3 children-10-01148-f003:**
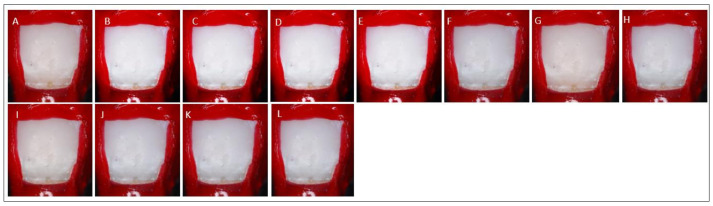
Image of the ROI at different stages: (**A**) Baseline, (**B**) Demineralization, (**C**) Etching cycle one, (**D**) Etching cycle two, (**E**) Etching cycle three, (**F**) Etching cycle four, (**G**) Etching cycle five, (**H**) Etching cycle six, (**I**) Etching cycle seven, (**J**) Etching cycle nine, (**K**) Resin 1, and (**L**) Resin 2.

**Table 1 children-10-01148-t001:** Descriptive analysis of surface roughness between different stages (baseline, etching, resin 1, resin 2, post resin 7 days, and post resin 28 days) and various etching groups.

Group	Stages (Mean ± SD)					
	Baseline	Etching	Resin 1	Resin 2	Post Resin 7 Days	Post Resin 28 Days
1	2.31 ± 0.064	3.64 ± 0.612	3.03 ± 0.664	1.53 ± 0.22	1.37 ± 0.195	1.34 ± 0.073
2	3.37 ± 0.225	4.37 ± 0.225	3.39 ± 0.492	1.54 ± 0.626	1.44 ± 0.569	1.49 ± 0.332
3	3.29 ± 0.153	4.96 ± 0.992	3.57 ± 0.216	1.74 ± 0.419	1.74 ± 0.467	1.74 ± 0.607
4	3.17 ± 0.737	5.51 ± 0.208	4.39 ± 0.367	1.95 ± 0.367	1.86 ± 0.467	1.96 ± 0.417
5	4.09 ± 0.429	6.21 ± 0.413	5.06 ± 0.724	2.54 ± 0.182	2.2 ± 0.049	2.62 ± 0.01
6	4.09 ± 1.364	6.31 ± 0.246	5.39 ± 0.63	2.67 ± 0.968	2.71 ± 0.968	2.64 ± 0.968
7	4.43 ± 0.238	6.44 ± 0.203	6.39 ± 0.358	2.87 ± 0.09	2.62 ± 0.395	2.81 ± 0.13
8	4.77 ± 0.737	6.56 ± 0.391	7.39 ± 0.196	2.44 ± 0.408	2.63 ± 0.075	2.26 ± 0.122
9	5.11 ± 0.537	6.59 ± 0.379	2.69 ± 0.313	3.44 ± 0.725	2.85 ± 0.458	2.77 ± 0.575

A Significant at *p*-value < 0.05, SD: Standard Deviation.

**Table 2 children-10-01148-t002:** Summary table related surface roughness for the interaction of stages and various etching group.

Source	SS	df	MS	f	*p*-Value
Interaction stages and etching group	46.334	48	0.965	3.48	<0.001 *
Stages	53.633	8	6.704	24.17	<0.001 *
Etching group	277.881	6	46.313	166.974	<0.001 *

SS = sum-of-square, MS = mean sum of squares, df = degrees of freedom. *p*-Value refers to two-way ANOVA. * A Significant at *p*-value < 0.05.

**Table 3 children-10-01148-t003:** Descriptive analysis of esthetic change between different stages (baseline, post etching, resin 1, resin 2, 7 days, and 28 days) and various etching groups. An increase in value is indicated by the color’s saturation.

Group	Stages (Mean ± SD)					
	Baseline	Etching	Resin 1	Resin 2	Post Resin 7 Days	Post Resin 28 Days
1	2.95 ± 0.06	3.64 ± 0.306	2.56 ± 0.905	1.83 ± 0.577	1.82 ± 0.811	1.72 ± 0.639
2	2.49 ± 0.81	4.37 ± 0.026	2.34 ± 0.279	1.74 ± 1.671	1.68 ± 0.446	1.72 ± 0.537
3	2.31 ± 0.104	4.96 ± 0.002	2.06 ± 0.329	1.68 ± 0.032	1.61 ± 0.272	1.69 ± 0.499
4	1.97 ± 0.012	5.51 ± 0.034	1.89 ± 0.974	1.63 ± 0.758	1.58 ± 0.375	1.67 ± 2.67
5	2.14 ± 1.209	6.21 ± 0.046	1.71 ± 0.172	1.58 ± 0.117	1.55 ± 0.484	1.64 ± 0.23
6	2.04 ± 0.353	6.51 ± 0.06	1.56 ± 0.095	1.53 ± 0.608	1.51 ± 0.305	1.64 ± 0.029
7	2.27 ± 0.196	6.52 ± 1.306	1.65 ± 0.057	1.48 ± 0.305	1.46 ± 0.703	1.54 ± 0.029
8	2.5 ± 0.433	6.53 ± 0.007	1.56 ± 0.046	1.44 ± 0.561	1.4 ± 0.34	1.14 ± 0.573
9	1.89 ± 0.234	6.54 ± 0.052	1.34 ± 0.833	1.3 ± 0.23	1.32 ± 0.499	1.19 ± 1.508

SD: Standard Deviation.

**Table 4 children-10-01148-t004:** A two-way ANOVA summary table related ΔE for the interaction of stages and various etching group.

Source	SS	df	MS	f	*p*-Value
Interaction stages and etching group	46.334	4	0.965	1.177	0.045 *
Stages	53.974	8	6.704	12.32	<0.001 *
Etching group	24.17	6	46.313	122.17	<0.001 *

SS = sum-of-square, MS = mean sum of squares, df = degrees of freedom. *p*-value refers to two-way ANOVA. * Significant at *p*-value < 0.05.

**Table 5 children-10-01148-t005:** Pairwise comparisons using Posthoc Tukey’S HSD Test (*p* < 0.05) for ΔE on different etching cycles.

Etching Cycle (I)	Etching Cycle (J)	Mean (I − J)	Std. Error	*p*-Value
1	2	3.557	0.272	<0.0001 *
	3	5.162	0.272	<0.0001 *
	4	3.883	0.272	<0.0001 *
	5	3.751	0.272	<0.0001 *
	6	5.952	0.27	<0.0001 *
	7	5.717	0.272	<0.0001 *
	8	6.345	0.272	<0.0001 *
	9	6.908	0.272	<0.0001 *
2	3	1.605	0.272	0.001 *
	4	0.327	0.272	<0.0001 *
	5	0.194	0.272	<0.0001 *
	6	2.395	0.272	<0.0001 *
	7	2.161	0.27	<0.0001 *
	8	2.789	0.272	<0.0001 *
	9	3.352	0.272	<0.0001 *
3	4	−1.278	0.272	0.007 *
	5	−1.411	0.272	0.002 *
	6	0.79	0.272	<0.0001 *
	7	0.555	0.272	<0.0001 *
	8	1.183	0.272	<0.0001 *
	9	1.746	0.272	<0.0001 *
4	7	1.834	0.272	<0.0001 *
	8	2.462	0.272	<0.0001 *
	9	3.025	0.272	<0.0001 *
5	1	1.966	0.272	<0.0001 *

* Significant at *p*-value < 0.05, Std. Error: Standard Error.

## Data Availability

The datasets generated during analyzed during the current study are available from the corresponding author on reasonable request.

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
