# Peer review of "Repeated Etching Cycles of Resin Infiltration up to Nine Cycles on Demineralized Enamel: Surface Roughness and Esthetic Outcomes—In Vitro Study"

_children, 2023, doi:10.3390/children10071148_

Round 1

Reviewer 1 Report

Dear Authors,

It is an interesting topic. However, some minor points need to be changed before considering it for publication:

2.         Materials and Methods

Line 107: Please erase “also”…leave only: “were excluded”

4.         Discussion

What are the strengths of your study? Can you find justification for a large number of etching cycles to get better surface characteristics and color stability?

Reviewer 2 Report

Title

Please add study design in your title.

Abstract

The abstract should be a single paragraph and should follow the style of structured abstracts, but without headings. Please check journal guidelines.

Line 16: “demineralized enamel lesions”. Not clear, either you say enamel lesions or demineralized enamel.

Please add reason for extracting ninety sound premolars.

Please add abbreviation following full term when possible. The first time an abbreviation appears, it should be placed in parentheses following the full spelling of the term.

Please use abbreviation (RI) for resin infiltration.

According to what, the teeth were divided into 9 groups?

Line 18: “9 groups”. Do you have “experimental and control groups”. Please clarify.

P-value not p-value.

The conclusion is lengthy, please reduce.

Keywords: check that they are all registered on MeSH.

Introduction

Line 60: it would be better if you start with "Incipient decay can now be managed with many conservative methods such as resin infiltration, which was developed in Germany."

Line 63: “demineralized enamel lesions”. Not clear, either you say enamel lesions or demineralized enamel.

Line 68: A recent systematic reviews. Please revise language.

Line 79: “number of etching cycles following repeated etching cycles”. Please avoid repetition.

Resin infiltration has several benefits, please mention them more clearly.

The Introduction does not give a rationale why this study should be conducted or what scientific value it has.

At the end of the Intro section, please give your null hypothesis. The latter should be derived from the preceding thoughts in this section and should be broached again in the Discussion. In hypothesis testing, the null hypothesis is the one you are hoping that is can be disproven by the observed data.

Materials and Methods

What is the study rationale?

2.1. Study design and sample preparation. This section did not mention the study design clearly.

Do you have “experimental and control groups”. Please clarify.

Where was the collected human premolars stored after extraction.

2.2. Artificial demineralization. Where the specimen was immersed individually in a separate glass that contained the demineralizing solution, please clarify.

You need to say, “After demineralization, the teeth were taken out of the demineralizing solution and washed with deionized water, dried”.

Why the teeth were not evaluated visually using the “International Caries Detection and Assessment System” “ICDAS” codes to make sure that only the teeth with codes 1 and 2 (early stage decay) were included.

Manufacturer names for all materials and equipment need to be included. e.g. “line 143: profilometer and spectrophotometer scans”.

“seven evaluation stages (baseline, post etching, resin 1, resin 2, post resin 7 days and post resin 28 days)”. I can see 6 only, please revise.

Why color measurements were taken after post resin 7 days and post resin 28 days?

2.4. Surface roughness. Please add a reference to this section.

Please use full term for “SPSS” followed by its abbreviation.

Please add abbreviation following full term when possible. The first time an abbreviation appears, it should be placed in parentheses following the full spelling of the term.

P-value not p-value.

Results

You need to explain more about your results not just write about what is in the tables and figures.

Please add calculated P-values. Please write P-value in italics (P-value).

(Refer to Table), please delete “Refer to”.

Lines 205 and 227: Please remove (*) from the text.

Tables and Figures

In all Tables and 3, “* A Significant at p-value <0.05”. There are no P -values in Tables 1 and 3.

Also, please add description to the groups, not just numbers.

In Tables 2 and 4, a footnote explaining the abbreviations (MS, f) needs to be added (what do they stands for). Also, level of significance needs to be mentioned.

In Tables 2 and 4, the legends are not clear. Please remove (*) from the text and the legend as well.

Discussion

This section may usefully start with a brief summary of the major findings.

The Discussion lacks continuity, most paragraphs just summarized other studies about resin infiltration and compared them to their results. The authors reported facts but did not give sufficient interpretation.

The long-term survival of resin infiltration was not discussed.

Please modify to “The current or present study” NOT “our study or this study”.

Resin infiltration increases surface roughness. The authors say resin infiltration can still lead to plaque accumulation. Does resin infiltration attract more plaque than demineralized lesions?

Please add high cost of RI in your limitations.

Please mention future directions.

Please point out the implications of the findings and their limitations.

Conclusions

Lengthy.

They are more related to results than conclusions.

 “In order to examine the relationship between impacted maxillary canines and malocclusion, further studies investigating the etiologic factors that cause impacted canines are needed”. This is a recommendation.

References

References needs to be 10 years back not more (from 2012 to 2022). Old references need to be replaced by recent ones.

Regarding references with no page range. You need to add DOI number.

Some references have missing information.

In general, all references need to be revised, standardized and written according to the journal guidelines.

In General

This is an in vitro study on extracted premolars. It studied the effect of resin infiltration on demineralized enamel after being embedded in mineralizing solutions. The authors name this demineralization "…induced white spot lesions ". Naturally this is not the case in the mouth where conditions are different and a large number of factors are involved, and saliva contamination needs to be taken care of. At this stage, needed are in vivo studies.

Minor editing of English language required

Reviewer 3 Report

The paper "Repeated etching cycles of resin infiltration on demineralized  enamel: Surface roughness and esthetic outcomes" deals with an important subject. However, the authors should be more specific: how many ethcing cycles? As well to be mentioned in the title as well as in the abstract. 

The abstract iss too long, it should be better organized. The methodology should also be better explained, it is confusing how many etching cycles each group had.

The resins 1 and 2 should be specified: what types of material they are. 

It is innteresting to mention what could happen after 28 days with surface roughness and color changes. 

What do you mean by dual application of resin infiltration? 

"Their clinical relevance must be elucidated through additional research" is not a conclusion.

In the introduction part it would be important to mention more details about resin infiltration: types of resin.

In the discussion part be more precise: 

Our data suggested that resin infiltration treatment decreases surface roughness relative to baseline similar to a recent meta-analysis by Soveral et al. (2021)" - how?

Figures and tables should be inserted in the manuscript, not at the end of the paper

Minor editing of English language should be advisable

Reviewer 4 Report

Excellent work! 

My only suggestion to analyze the synthesis and characteristics of infiltration resin(to discussion part). If possible also try to explain why this methods uses HCL rather than ortho-phosphoric acid. Clinically very helpful study!

Round 2

Reviewer 2 Report

none

Reviewer 3 Report

The manuscript has been improved